



# Ensemble Riemannian Data Assimilation over the Wasserstein Space

Sagar K. Tamang[1], Ardeshir Ebtehaj[1], Peter J. van Leeuwen[2], Dongmian Zou[3], and Gilad Lerman[4]

[1]Department of Civil, Environmental and Geo-Engineering and Saint Anthony Falls Laboratory, University of Minnesota-Twin Cities, Twin Cities, Minnesota, USA

[2]Department of Atmospheric Science, Colorado State University, Fort Collins, Colorado, USA

[3]Duke Kunshan University, Kunshan, China

[4]School of Mathematics, University of Minnesota-Twin Cities, Twin Cities, Minnesota, USA

**Correspondence:** Sagar K. Tamang (taman011@umn.edu), Ardeshir Ebtehaj (ebtehaj@umn.edu)

**Abstract.** In this paper, we present an ensemble data assimilation paradigm over a Riemannian manifold equipped with the Wasserstein metric. Unlike the Eulerian penalization of error in the Euclidean space, the Wasserstein metric can capture translation and difference between the shapes of square-integrable probability distributions of the background state and observations – enabling to formally penalize geophysical biases in state-space with non-Gaussian distributions. The new approach is applied

to dissipative and chaotic evolutionary dynamics and its potential advantages and limitations are highlighted compared to the classic variational and filtering data assimilation approaches under systematic and random errors.

## 1 Introduction

Extending the forecast skill of Earth System Models (ESM) relies on advancing the science of Data Assimilation (DA) (Tsuyuki and Miyoshi, 2007; Carrassi et al., 2018). A large body of current DA method-

ologies, either filtering (Kalman, 1960; Evensen, 1994; Reichle et al., 2002; Evensen, 2003) or variational approaches (Lorenc, 1986; Le Dimet and Talagrand, 1986; Talagrand and Courtier, 1987; Park and Županski, 2003; Trevisan et al., 2010; Carrassi and Vannitsem, 2010; Ebtehaj and Foufoula-Georgiou, 2013), are derived from basic principles of Bayesian inference under the assumption that the state-space is unbiased and can be represented well with Gaussian distributions, which are not often consistent with reality

(Bocquet et al., 2010; Pires et al., 2010). It is well documented that this drawback often limits forecast skills of DA systems (Walker et al., 2001; Dee, 2005; Ebtehaj et al., 2014; Chen et al., 2019a) especially under the presence of systematic errors (Dee, 2003).

Apart from particle filters (Spiller et al., 2008; Van Leeuwen, 2010), which are intrinsically designed for state-space with non-Gaussian distribution, numerous modifications to the variational DA (VDA) and

ensemble-based filtering methods have been made to tackle non-Gaussianity of geophysical processes



(Pires et al., 1996; Han and Li, 2008; Mandel and Beezley, 2009; Anderson, 2010). As a few examples, in four-dimensional VDA, a quasi-static VDA is proposed to ensure convergence by gradually increasing the assimilation intervals (Pires et al., 1996). Kim et al. (2003) proposed modifications to the ensemble Kalman filter (EnKF; Evensen, 1994; Li et al., 2009) using approximate implementation of Bayes' theo-

rem in lieu of linear interpolation via Kalman gain to deal with multimodal systems. For ensemble-based filters, Anderson (2010) proposed a new approach to account for non-Gaussian priors and posteriors by utilizing rank histograms (Anderson, 1996; Hamill, 2001). A hybrid ensemble approach was also suggested to combine advantages of both EnKF and particle filter (Mandel and Beezley, 2009).

Even though particle filters can handle non-Gaussian likelihood functions, when observations lie away

from the support set of the particles, the ensemble variance tends to zero over time and can render the filter degenerate (Poterjoy and Anderson, 2016). In recent years, significant progress has been made to treat systematic errors through numerous ad hoc methods such as the field alignment technique (Ravela et al., 2007) and morphing EnKF (Beezley and Mandel, 2008) that can tackle position errors between observations and forecast. Dual state-parameter EnKF (Moradkhani et al., 2005) was also developed to

resolve systematic errors originating from parameter uncertainties. Additionally, bias aware variants of the Kalman filter were designed (Drécourt et al., 2006; De Lannoy et al., 2007a, b; Kollat et al., 2008) to simultaneously update the state-space and an *a priori* estimate of the additive biases. In parallel, the cumulative distribution function matching (Reichle and Koster, 2004) has garnered widespread attention in land DA.

From a geometrical perspective, Gaussian statistical inference methods exhibit a flat geometry (Amari, 1985). In particular, it is proved that linear auto-regressive and moving average Markov stochastic models, which are driven by Gaussian noise, form dually flat manifolds (Amari, 2012). The notion of distance over such a geometrically flat space is defined over a straight line, which can be quantified by the Euclidean distance. Consequently, the Euclidean space has served as a major tool in explaining statistical inference

techniques using linear Gaussian models and has been used as a cornerstone of DA techniques. It is important to note that the Euclidean distance is "Eulerian" (Ning et al., 2014) and thus remains insensitive to the magnitude of translation between probability distributions with disjoint support sets – when used to interpolate between them.

Non-Gaussian statistical models often form geometrical manifolds. In the case of nonlinear regression,

it is demonstrated that the statistical manifold exhibits a Riemannian geometry (Lauritzen, 1987) over





which the notion of distance between probability distributions is geodesic. Such a distance metric shall be Lagrangian to not only capture translation but also the difference between the entire shape of probability distributions (Pennec, 2006). How can we equip DA with a Riemannian geometry? To answer this question, inspired by the theories of optimal mass transport (Villani, 2003), this paper presents the Ensemble

Riemannian Data Assimilation (EnRDA) framework using the Wasserstein distance metric.

In recent years, a few attempts have been made to utilize the Wasserstein metric in geophysical DA. Reich (2013) introduced an ensemble transform particle filter, where the optimal transport framework was utilized to guide the resampling phase of the filter. Ning et al. (2014) used the Wasserstein distance to reduce forecast uncertainty due to parameter estimation errors in dissipative evolutionary equations. Feyeux

et al. (2018) suggested a novel approach employing the Wasserstein distance in lieu of the Euclidean distance to penalize the position error between state and observations. More recently, Tamang et al. (2020) introduced a Wasserstein regularization in a variational setting to correct for geophysical biases under chaotic dynamics.

The EnRDA extends the previous work through the following main contributions: (a) EnRDA defines

DA as a discrete barycenter problem over the Wasserstein space for assimilation in probability domain without any parametric or Gaussian assumption. The framework provides a continuum of non-parametric analysis probability histograms that naturally span between the distributions of the background state and observations through optimal transport of probability masses. (b) EnRDA operates in an ensemble setting using the entropic regularization by utilizing the Sinkhorn algorithm (Cuturi, 2013) for improving com-

putational efficiency. (c) The paper studies advantages and limitations of DA over the Wasserstein space for dissipative advection-diffusion dynamics and nonlinear chaotic Lorenz-63 model in comparison with 3D Variational (3D-Var) DA as well as filtering techniques.

The organization of the paper is as follows: Section 2 provides a brief background on Bayesian DA formulations and optimal mass transport. The mathematical formalism of the EnRDA is described in

Section 3. Section 4 presents the results and compares them with their Euclidean counterparts. Section 5 discusses the findings and ideas for future research.





## 2   Background

### 2.1   Notations

Throughout, small bold letters represent $m$-element column vectors $\mathbf{x} = (x_1, \ldots, x_m)^{\mathrm{T}} \in \mathbb{R}^m$, where $(\cdot)^{\mathrm{T}}$
is the transposition operator. The $m$-by-$n$ matrices $\mathbf{X} \in \mathbb{R}^{m \times n}$ are denoted by capital bold letters, whereas
$\mathbb{R}_+^m (\mathbb{R}_+^{m \times n})$ denotes those vectors (matrices) only containing non-negative real numbers. The $\mathbb{1}_m$ refers to
an $m$-element vector of ones and $\mathbf{I}_m$ is an $m \times m$ identity matrix. A diagonal matrix with entries given
by $\mathbf{x} \in \mathbb{R}^m$ is represented by $\mathrm{diag}(\mathbf{x}) \in \mathbb{R}^{m \times m}$. Notation $\mathbf{x} \sim \mathcal{N}(\boldsymbol{\mu}, \boldsymbol{\Sigma})$ denotes that the random vector
$\mathbf{x}$ is drawn from a Gaussian distribution with mean $\boldsymbol{\mu}$ and covariance $\boldsymbol{\Sigma}$ and $\mathbb{E}_X(\mathbf{x})$ is the expectation
of $\mathbf{x}$. The $\ell_q$-norm of $\mathbf{x}$ is defined as $\|\mathbf{x}\|_q = \left( \sum_{i=1}^m |x_i|^q \right)^{1/q}$ with $q > 0$ and the square of the weighted
$\ell_2$-norm of $\mathbf{x}$ is represented as $\|\mathbf{x}\|_{\mathbf{B}^{-1}}^2 = \mathbf{x}^{\mathrm{T}} \mathbf{B}^{-1} \mathbf{x}$, where $\mathbf{B}$ is a positive definite matrix. Notations of
$\mathbf{x} \odot \mathbf{y}$ and $\mathbf{x} \oslash \mathbf{y}$ represent the element-wise Hadamard product and division between equal length vectors
$\mathbf{x}$ and $\mathbf{y}$, respectively. Notation $\langle \mathbf{A}, \mathbf{B} \rangle = \mathrm{tr}(\mathbf{A}^{\mathrm{T}} \mathbf{B})$ denotes the Frobenius inner product between matrices
$\mathbf{A}$ and $\mathbf{B}$ and $\mathrm{tr}(\cdot)$ and $\det[\cdot]$ represent trace and determinant of a square matrix, respectively. Here, $p(\mathbf{x}) = \sum_{i=1}^M p_{\mathbf{x}_i} \delta_{\mathbf{x}_i}$ represents a discrete probability distribution with respective histogram $\{\mathbf{p}_x \in \mathbb{R}_+^M : \sum_i p_{\mathbf{x}_i} = 1\}$ supported on $\mathbf{x}_i$, where $\delta_{\mathbf{x}_i}$ represents a Kronecker delta function at $\mathbf{x}_i$. Throughout, the dimension
of the state or observations is denoted by little letters such as $\mathbf{x} \in \mathbb{R}^m$ while the number of ensembles or
support points of their respective probability distribution is shown by capital letters such as $\mathbf{p}_x \in \mathbb{R}_+^M$.

### 2.2   Data Assimilation on Euclidean Space

In this section, we provide a brief review of the derivation of classic variational DA and particle filters
based on the Bayes' theorem to set the stage for the presented Ensemble Riemannian DA formalism.

#### 2.2.1   Variational Formulation

Let us consider a discrete-time Markovian dynamics and its observations as follows:

$$
\begin{aligned}
\mathbf{x}^t &= \mathcal{M}(\mathbf{x}^{t-1}) + \boldsymbol{\omega}^t, & \boldsymbol{\omega}^t &\sim \mathcal{N}(\mathbf{0}, \mathbf{B}) \\
\mathbf{y}^t &= \mathcal{H}(\mathbf{x}^t) + \boldsymbol{v}^t, & \boldsymbol{v}^t &\sim \mathcal{N}(\mathbf{0}, \mathbf{R}),
\end{aligned}
\tag{1}
$$





where $\mathbf{x}^t \in \mathbb{R}^m$ and $\mathbf{y}^t \in \mathbb{R}^n$ represent the state variables and the observations at time $t$, $\mathcal{M} : \mathbb{R}^m \to \mathbb{R}^m$
and $\mathcal{H} : \mathbb{R}^m \to \mathbb{R}^n$ are the deterministic forward model and observation operator, and $\boldsymbol{\omega}^t \in \mathbb{R}^m$ and $\boldsymbol{v}^t \in$
$\mathbb{R}^n$ are the independent and identically distributed model and observation errors, respectively.

Recalling the Bayes' theorem, dropping the time superscript, without loss of generality, the poste-
rior probability density function (pdf) of the state given the observation can be obtained as $p(\mathbf{x}|\mathbf{y}) \propto$
$p(\mathbf{y}|\mathbf{x})\,p(\mathbf{x})/p(\mathbf{y})$, where $p(\mathbf{y}|\mathbf{x})$ is proportional to the likelihood function, $p(\mathbf{x})$ is the prior density and
$p(\mathbf{y})$ denotes the distribution of observations. Letting $\mathbf{x}_b = \mathbb{E}_X(\mathbf{x}) \in \mathbb{R}^m$ represents the background state,
ignoring the constant term $\log p(\mathbf{y})$ and assuming Gaussian distributions for the observation error and the
prior, logarithm of the posterior density leads to the known three-dimensional variational (3D-Var) cost
function (Lorenc, 1986):

$$
\begin{aligned}
-\log p(\mathbf{x}|\mathbf{y}) &\propto \frac{1}{2}(\mathbf{x} - \mathbf{x}_b)^{\mathrm{T}} \mathbf{B}^{-1}(\mathbf{x} - \mathbf{x}_b) + \frac{1}{2}(\mathbf{y} - \mathcal{H}(\mathbf{x}))^{\mathrm{T}} \mathbf{R}^{-1}(\mathbf{y} - \mathcal{H}(\mathbf{x})) \\
&\propto \|\mathbf{x} - \mathbf{x}_b\|_{\mathbf{B}^{-1}}^2 + \|\mathbf{y} - \mathcal{H}(\mathbf{x})\|_{\mathbf{R}^{-1}}^2 .
\end{aligned}
\tag{2}
$$

As a result, the analysis state obtained by minimization of the 3D-Var cost function in Eq. (2) is the mode
of the posterior distribution that coincides with the posterior mean when errors are drawn from unbiased
Gaussian densities and $\mathcal{H}$ is a linear operator. Using the Woodbury matrix inversion lemma (Woodbury,
1950), it can be easily demonstrated that for a linear observation operator, the analysis states in the 3D-Var
and Kalman filter are equivalent (Tarantola, 1987). As is evident, zero-mean Gaussian assumptions lead
to penalization of the error through the weighted Euclidean norm.

### 2.2.2    Particle Filters

Particle filters (Gordon et al., 1993; Doucet and Johansen, 2009; Van Leeuwen et al., 2019) in DA were
introduced to address the issue of non-Gaussian distribution of the state by representing the prior and
posterior distributions through a weighted ensemble of model outputs referred to as "particles". In its
standard discrete setting, using Monte Carlo simulations, the prior distribution $p(\mathbf{x})$ is represented by a
sum of equal-weight Kronecker delta functions as $p(\mathbf{x}) = \dfrac{1}{M}\displaystyle\sum_{i=1}^{M}\delta_{\mathbf{x}_i}$, where $\mathbf{x}_i \in \mathbb{R}^m$ is the state variable
represented by the $i^{\mathrm{th}}$ particle.





Each of these $M$ particles are then evolved through the nonlinear model in Eq. (1). Assuming that the
conditional distribution $p(\mathbf{y}|\mathbf{x}_i) = \frac{1}{(2\pi)^{n/2}|\mathbf{R}|^{1/2}}\exp\left\{-\frac{1}{2}[\mathbf{y}-\mathcal{H}(\mathbf{x}_i)]^\mathrm{T}\mathbf{R}^{-1}[\mathbf{y}-\mathcal{H}(\mathbf{x}_i)]\right\}$ is Gaussian,
using the Bayes' theorem, it can be shown that the posterior distribution $p(\mathbf{x}|\mathbf{y})$ can be approximated
using a set of weighted particles as $p(\mathbf{x}|\mathbf{y}) = \sum_{i=1}^{M} w_i \delta_{\mathbf{x}_i}$, where $w_i = \frac{p(\mathbf{y}|\mathbf{x}_i)}{\sum_{j=1}^{M} p(\mathbf{y}|\mathbf{x}_j)}$. The particles are
then resampled from the posterior distribution $p(\mathbf{x}|\mathbf{y})$ based on their relative weights and propagated
forward in time according to the model dynamics.

As is evident, in particle filters, weights of each particle are updated using the Gaussian likelihood func-
tion under a zero-mean error assumption. However, in the presence of systematic biases, when the support
sets of particles and the observations are disjoint, only the weights of a few particles become significantly
large and weights of other particles tend to zero. As the underlying dynamical system progresses in time,
only those few particles, with relatively larger weights, are resampled and the filter can become degenerate
gradually in time (Poterjoy and Anderson, 2016).

### 2.3 Optimal Mass Transport

The theory of optimal mass transport (OMT), coined by Gaspard Monge (Monge, 1781) and later extended
by Kantorovich (Kantorovich, 1942), was developed to minimize transportation cost in resource allocation
problems with purely practical motivations. Recent developments in mathematics discovered that OMT
provides a rich ground to compare and morph probability distributions and uncovered new connections
to partial differential equations (Jordan et al., 1998; Otto, 2001) and functional analysis (Brenier, 1987;
Benamou and Brenier, 2000; Villani, 2003).

In a discrete setting, let us define two discrete probability distributions $p(\mathbf{x}) = \sum_{i=1}^{M} p_{\mathbf{x}_i}\delta_{\mathbf{x}_i}$ and $p(\mathbf{y}) = \sum_{j=1}^{N} p_{\mathbf{y}_j}\delta_{\mathbf{y}_j}$ with their respective histograms $\{\mathbf{p}_x \in \mathbb{R}_+^M : \sum_i p_{\mathbf{x}_i} = 1\}$ and $\{\mathbf{p}_y \in \mathbb{R}_+^N : \sum_j p_{\mathbf{y}_j} = 1\}$ sup-
ported on $\mathbf{x}_i$ and $\mathbf{y}_j$. A "ground" transportation cost matrix $\mathbf{C} \in \mathbb{R}_+^{M \times N}$ is defined such that its elements
$c_{ij} = \|\mathbf{x}_i - \mathbf{y}_j\|_q^q$ represent the cost of transporting unit probability masses from location $\mathbf{x}_i$ to $\mathbf{y}_j$. The
Kantorovich OMT problem determines an optimal "transportation plan" $\mathbf{U}^a \in \mathbb{R}_+^{M \times N}$ that can linearly
map two probability measures onto each other with minimum amount of total transportation cost as fol-
lows:

$$\mathbf{U}^a = \operatorname*{argmin}_{\mathbf{U}} \langle \mathbf{C}, \mathbf{U} \rangle \qquad \text{s.t.} \qquad \mathbf{U} \geq 0, \quad \mathbf{U}\mathbb{1}_N = \mathbf{p}_x, \quad \mathbf{U}^\mathrm{T}\mathbb{1}_M = \mathbf{p}_y. \tag{3}$$





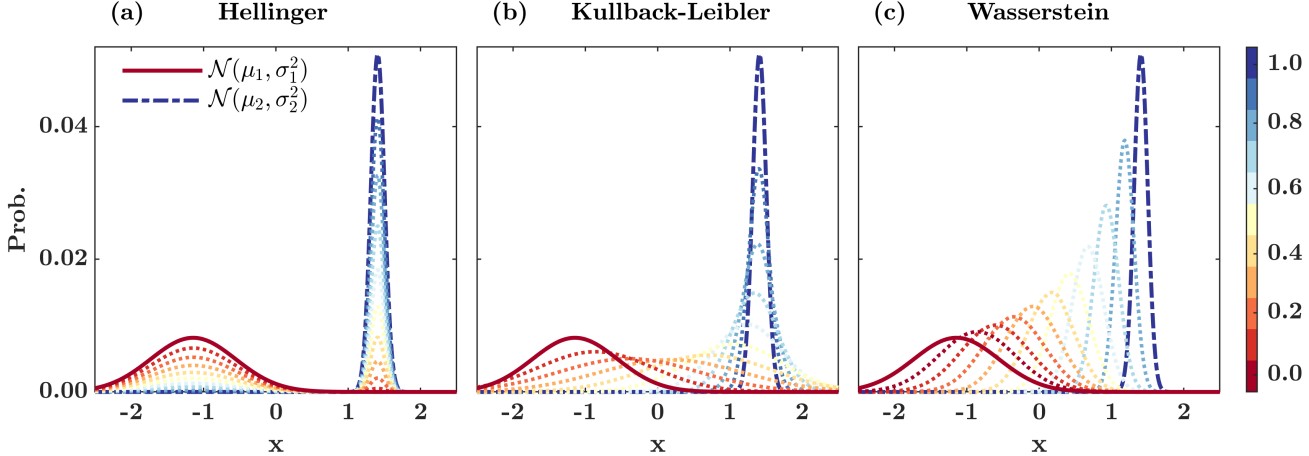

**Figure 1.** Interpolation between two Gaussian distributions $\mathcal{N}(\mu_1, \sigma_1^2)$ and $\mathcal{N}(\mu_2, \sigma_2^2)$ where, $\mu_1 = -1.1, \mu_2 = 1.4, \sigma_1^2 = 0.4$, and $\sigma_2^2 = 0.01$ as a function of displacement parameter $\eta \in [0, 1]$ for the (a) Hellinger distance, (b) Kullback-Leibler divergence, and (c) 2-Wasserstein distance (Peyré and Cuturi, 2019).

The transportation plan can be interpreted as a "joint distribution" that couples the marginals histograms $\mathbf{p}_x$ and $\mathbf{p}_y$. For the transportation cost with $q = 2$, the OMT problem in Eq. (3) is convex and defines the square of the 2-Wasserstein distance between the distributions as $d_{\mathcal{W}}^2(\mathbf{p}_x, \mathbf{p}_y) = \langle \mathbf{C}, \mathbf{U^a} \rangle$.

What is the advantage of the Wasserstein distance for interpolating between probability distributions

compared to other measures of proximity – such as the Hellinger distance (Hellinger, 1909) or the Kullback–Leibler (KL) divergence (Kullback and Leibler, 1951)? To elaborate on this question, we confine our consideration to the Gaussian densities over which the Wasserstein distance can be represented in a closed form. In particular, interpolating over the 2-Wasserstein space using parameter $\eta$, between $\mathcal{N}(\boldsymbol{\mu}_x, \boldsymbol{\Sigma}_x)$ and $\mathcal{N}(\boldsymbol{\mu}_y, \boldsymbol{\Sigma}_y)$, results in a Gaussian distribution $\mathcal{N}(\boldsymbol{\mu}_\eta, \boldsymbol{\Sigma}_\eta)$, where $\boldsymbol{\mu}_\eta = \eta\,\boldsymbol{\mu}_x + (1-\eta)\,\boldsymbol{\mu}_y$

and $\boldsymbol{\Sigma}_\eta = \boldsymbol{\Sigma}_x^{-1/2}\big(\eta\,\boldsymbol{\Sigma}_x + (1-\eta)\,(\boldsymbol{\Sigma}_x^{1/2}\boldsymbol{\Sigma}_y\boldsymbol{\Sigma}_x^{1/2})^{1/2}\big)^2 \boldsymbol{\Sigma}_x^{-1/2}$ (Chen et al., 2019b).

Fig. 1 shows the spectrum of interpolated distributions between two Gaussian pdfs for a range of the interpolation parameter $\eta \in [0, 1]$. As shown, the interpolated densities using the Hellinger distance, which is Euclidean in the space of probability measure, are bimodal. Although the Gaussian shape of the interpolated densities using the KL divergence is preserved, the variance of the interpolants is not necessarily

bounded by the variances of the input Gaussian densities. Unlike these metrics, as shown, the Wasserstein distance moves the mean and preserves the shape of the interpolants through a natural morphing process.

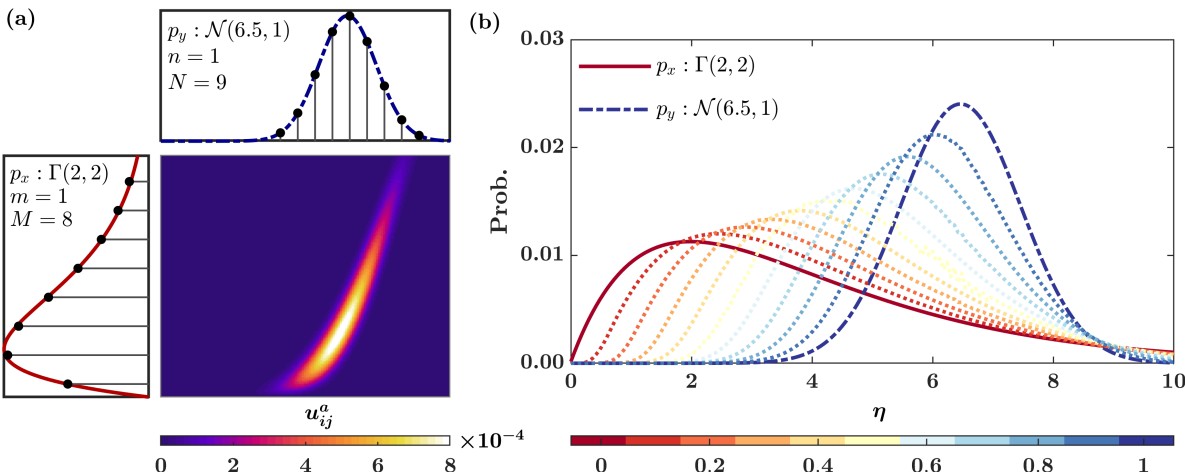

**Figure 2.** (a) Optimal transportation plan or the joint distribution $\mathbf{U}^a$ between a gamma $\Gamma(2,2)$ and a Gaussian marginal distribution $\mathcal{N}(6.5,1)$ as well as (b) the 2-Wasserstein interpolation between them for different values of the displacement parameter $\eta \in [0,1]$.

As is previously noted, this metric is not limited to any Gaussian assumption. Fig. 2 shows the 2-Wasserstein interpolation between a gamma and a Gaussian distribution. The results show the Lagrangian nature of the Wasserstein metric that penalizes the translation and mismatch between the shapes of the pdfs. It can be shown that $d_{\mathcal{W}}^2(\mathbf{p}_x, \mathbf{p}_y) = d_{\mathcal{W}}^2(\overline{\mathbf{p}}_x, \overline{\mathbf{p}}_y) + \left\| \boldsymbol{\mu}_x - \boldsymbol{\mu}_y \right\|_2^2$, where $\overline{\mathbf{p}}_x$ and $\overline{\mathbf{p}}_y$ are the centered zero-mean probability masses and $\boldsymbol{\mu}_x$ and $\boldsymbol{\mu}_y$ are the respective mean values (Peyré and Cuturi, 2019).

## 3 Ensemble Riemannian Data Assimilation

### 3.1 Problem Formulation

First, let us recall that the weighted mean of a cloud of points $\{\mathbf{x}_i\}_{i=1}^M \in \mathbb{R}^m$ in the Euclidean space is $\boldsymbol{\mu}_x = \sum_{i=1}^M \eta_i \mathbf{x}_i / M$ for a given family of non-negative weights $\sum_i \eta_i = 1$. This expected value is equivalent to solving the following variational problem:

$$\boldsymbol{\mu}_x = \operatorname*{argmin}_{\mathbf{x}} \sum_{i=1}^M \eta_i \left\| \mathbf{x}_i - \mathbf{x} \right\|_2^2 . \tag{4}$$

Thus, the 3D-Var problem in Eq. (2), after Cholesky decomposition (Nash, 1990) of the error covariance matrices and rearrangement of the terms, can be interpreted as a "barycenter problem" in the Euclidean space, where the analysis state is the weighted mean of the background state and observation.





By changing the distance metric from Euclidean to the Wasserstein (Agueh and Carlier, 2011), a Riemannian barycenter can be defined as the Fréchet mean (Fréchet, 1948) of $N_p$ probability histograms with finite second-order moments as follows:

$$\mathbf{p}_\eta = \underset{\mathbf{p}}{\mathrm{argmin}} \sum_{k=1}^{N_p} \eta_k \, d_{\mathcal{W}}^2(\mathbf{p}, \mathbf{p}_k). \tag{5}$$

Inspired by (Feyeux et al., 2018), the EnRDA defines the probability distribution of the analysis state $p(\mathbf{x}_a) \in \mathbb{R}^M$ as the Fréchet barycenter over the Wasserstein space as follows:

$$p(\mathbf{x}_a) = \underset{\mathbf{p}_x}{\mathrm{argmin}} \left\{ \eta \, d_{\mathcal{W}}^2(\mathbf{p}_x, \mathbf{p}_{x_b}) + (1-\eta) \, d_{\mathcal{W}}^2(\mathbf{p}_x, |\det[\mathcal{H}'(\mathbf{x})]| \, \mathbf{p}_y) \right\}, \tag{6}$$

where the displacement parameter $\eta > 0$ assigns the relative weights to the observation and background term to capture their respective geodesic distances from the true state. Here $\mathcal{H}'(\cdot)$ is the Jacobian of the
observation operator assuming that $\mathcal{H} : \mathbf{x} \to \mathbf{y}$ is a smooth and a square (i.e., $m = n$) bijective map. The $\eta$ is a hyperparameter and its optimal value should be determined empirically using some reference data through cross-validation studies. It is also important to note that due to the bijective assumption for the observation operator, the above formalism currently lacks the ability to propagate the information content of observed dimensions to unobserved ones. This limitation is further discussed later on in the section 5.
The solution of the above DA formalism involves finding the optimal analysis transportation plan or the joint distribution $\mathbf{U}^a \in \mathbb{R}^{M \times N}$, using Eq. (3), which couples the background and observation marginal histograms. From the joint histogram $\mathbf{U}^a$, we use the McCann's method (McCann, 1997; Peyré et al., 2019) to obtain the analysis probability distribution:

$$p(\mathbf{x}_a) = \sum_{i=1}^{M} \sum_{j=1}^{N} u_{ij}^a \, \delta_{\mathbf{z}_{ij}}, \tag{7}$$

where the analysis support points are $\mathbf{z}_{ij} = \eta \, \mathbf{x}_i + (1-\eta) \, \mathbf{y}_j$. The widely used interior-point methods (Altman and Gondzio, 1999) and the Orlin's (Orlin, 1993) algorithm which are used to solve Eq. (3), have super-cubic run time with a computational complexity of $O(M^3 \log M)$, where $M = N$. This is a limitation in high-dimensional geophysical DA problems that will be addressed in the next subsection.





To solve Eq. (6) in an ensemble setting, let us assume that in the absence of any a priori information,
initially the background probability distribution is represented by $i = 1 \ldots M$ ensemble members of the
state variable $\mathbf{x}_i \in \mathbb{R}^m$ as $p(\mathbf{x}_b) = \frac{1}{M} \sum_{i=1}^{M} \delta_{\mathbf{x}_i}$. An a priori assumption is needed to reconstruct the ob-
servation distribution $p(\mathbf{y}) = \sum_{i=1}^{N} \mathbf{p}_{y_j} \delta_{\mathbf{y}_j}$ at $j = 1 \ldots N$ supporting points. To that end, one may choose
a parametric or a non-parametric model based on the past climatological information. Here, for simplic-
ity, we assume a zero-mean Gaussian representation with covariance $\mathbf{R} \in \mathbb{R}^{n \times n}$ similar to the suggested
approach in (Burgers et al., 1998) that can be used to perturb the given observation at each assimilation cy-
cle. After each assimilation cycle, the probability histogram of the analysis state $p(\mathbf{x}_a)$ is recovered from
Eq. (8) over $\mathbf{z}_{ij}$ at $M \times N$ support points. Then $p(\mathbf{x}_a)$ is resampled at $M$ points using the multinomial
sampling scheme (Li et al., 2015) to initialize the next time step forecasts.

### 3.2 Entropic Regularization of EnRDA

In order to speed up the computation of coupling between $\mathbf{p}_{x_b}$ and $\mathbf{p}_y$, the problem in Eq. (3) can be
regularized (Cuturi, 2013) as follows:

$$\mathbf{U}^a = \underset{\mathbf{U}}{\mathrm{argmin}} \ \langle \mathbf{C}, \mathbf{U} \rangle - \gamma H(\mathbf{U}) \qquad \text{s.t. } \mathbf{U} \geq 0, \ \mathbf{U} \mathbb{1}_N = \mathbf{p}_{x_b}, \ \mathbf{U}^{\mathrm{T}} \mathbb{1}_M = \mathbf{p}_y, \tag{8}$$

where $\gamma > 0$ is the regularization parameter and $H(\mathbf{U}) := \langle \mathbf{U}, \log \mathbf{U} - \mathbb{1}_M \mathbb{1}_N^{\mathrm{T}} \rangle$ represents the Gibbs-
Boltzmann relative entropy function. Note that the relative entropy is a concave function and thus its
negative value is convex.

The Lagrangian function ($\mathcal{L}$) of Eq. (8) can be obtained by adding two dual variables or Lagrangian
multipliers $\mathbf{q} \in \mathbb{R}^M$ and $\mathbf{r} \in \mathbb{R}^N$ as follows:

$$\mathcal{L}(\mathbf{U}, \mathbf{q}, \mathbf{r}) = \langle \mathbf{C}, \mathbf{U} \rangle - \gamma H(\mathbf{U}) - \langle \mathbf{q}, \mathbf{U} \mathbb{1}_N - \mathbf{p}_{x_b} \rangle - \langle \mathbf{r}, \mathbf{U}^{\mathrm{T}} \mathbb{1}_M - \mathbf{p}_y \rangle. \tag{9}$$

Setting the derivative of the Lagrangian function to zero, we have

$$\frac{\partial \mathcal{L}(\mathbf{U}, \mathbf{q}, \mathbf{r})}{\partial u_{ij}} = c_{ij} + \gamma \log(u_{ij}) - q_i - r_j = 0, \qquad \forall i, j. \tag{10}$$





The convexity of the entropic regularization keeps the problem in Eq. (8) strongly convex and it can be shown (Peyré et al., 2019) that Eq. (10) leads to a unique optimal joint density with the following form:

$$\mathbf{U}^a = \mathrm{diag}(\mathbf{v})\,\mathbf{K}\,\mathrm{diag}(\mathbf{w})\,, \tag{11}$$

where $\mathbf{v} = \exp(\mathbf{q}) \oslash (\gamma \mathbb{1}_M) \in \mathbb{R}^M$ and $\mathbf{w} = \exp(\mathbf{r}) \oslash (\gamma \mathbb{1}_N) \in \mathbb{R}^N$ are the unknown scaling variables and $\mathbf{K} \in \mathbb{R}_+^{M \times N}$ is the Gibbs kernel, associated with cost matrix $\mathbf{C}$ with element $k_{ij} = \exp\left(-\frac{c_{ij}}{\gamma}\right)$.

From the mass conservation constraints in Eq. (8) and scaling form of the optimal joint density in Eq. (11), we can derive

$$\mathrm{diag}(\mathbf{v})\,\mathbf{K}\,\mathrm{diag}(\mathbf{w})\mathbb{1}_N = \mathbf{p}_{x_b} \qquad \text{and} \qquad \mathrm{diag}(\mathbf{w})\,\mathbf{K}^{\mathrm{T}}\,\mathrm{diag}(\mathbf{v})\mathbb{1}_M = \mathbf{p}_y\,. \tag{12}$$

The two unknown scaling variables $\mathbf{v}$ and $\mathbf{w}$ in Eq. (11) can be iteratively solved using the Sinkhorn's algorithm (Cuturi, 2013) as follows:

$$\mathbf{v}^{l+1} = \mathbf{p}_{x_b} \oslash (\mathbf{K}\mathbf{w}^l) \qquad \text{and} \qquad \mathbf{w}^{l+1} = \mathbf{p}_y \oslash (\mathbf{K}^{\mathrm{T}}\mathbf{v}^l)\,. \tag{13}$$

A summary of the EnRDA implementation is demonstrated in Algorithm 1.

The entropic regularization parameter $\gamma$ plays an important role in characterization of the joint density; however, there exists no closed-form solution for its optimal selection. Generally speaking, increasing the value of $\gamma$ will increase convexity of the cost function and thus computational efficiency; however, at the expense of reduced coupling between the marginal histograms, consistent with the second law of thermodynamics.

As an example, the effects of $\gamma$ on the coupling between two Gaussian mixture models $\mathbf{p}_{x_b}$ and $\mathbf{p}_y$ are demonstrated in Fig. 3. It can be seen that at smaller values of $\gamma = 0.001$, the probability masses of the joint distribution are sparse and lie compactly along the main diagonal – capturing a strong coupling between the background state and observations. However, as the value of $\gamma$ increases, the probability masses of the joint distribution spread out – reflecting less degree of dependencies between the marginals. It is important to note that in limiting cases, as $\gamma \to 0$, the solution of Eq. (8) converges to the true optimal joint histogram, while as $\gamma \to \infty$ the entropy of the analysis state increases and tends to $\mathbf{p}_{x_b}\mathbf{p}_y^{\mathrm{T}}$. Throughout, we empirically choose a minimum value for $\gamma$ that leads to a stable solution by the Sinkhorn





---

**Algorithm 1** Ensemble Riemannian Data Assimilation

---

1: **Inputs:** Ensemble size $M$, number of perturbed observations $N$ from a chosen observation pdf, displacement parameter $\eta$, entropic regularization parameter $\gamma$ and total number of time steps $T$.

2: **Initialize:** $\mathbf{x}_i^0 \sim p(\mathbf{x}^0), \ i = 1, \dots, M$.

3: **for** $t = 1, \dots, T$ **do**

4: $\quad \mathbf{x}_i^t = \mathcal{M}(\mathbf{x}_i^{t-1}) + \boldsymbol{\omega}_i^t, \ i = 1, \dots, M$.

5: $\quad$ Generating ensemble of observations $\mathbf{y}_j^t, \ j = 1, \dots, N$.

6: $\quad$ At initial time obtain probability histogram of the background state and observations:

$$p(\mathbf{x}_b) = \frac{1}{M} \sum_{i=1}^M \delta_{\mathbf{x}_i^t}, \quad p(\mathbf{y}) = \sum_{j=1}^N \mathbf{P}_{y_j} \delta_{\mathbf{y}_j^t}.$$

7: $\quad$ Compute the joint histogram as follows:

$$\mathbf{U}^a = \underset{\mathbf{U}}{\arg\min} \sum_{i=1}^M \sum_{j=1}^N u_{ij} c_{ij} - \gamma \langle \mathbf{U}, \log \mathbf{U} - \mathbb{1}_M \mathbb{1}_N^{\mathrm{T}} \rangle \quad \text{s.t. } \mathbf{U} \geq 0, \ \mathbf{U} \mathbb{1}_N = \mathbf{p}_{x_b}, \ \mathbf{U}^{\mathrm{T}} \mathbb{1}_M = \mathbf{p}_y,$$

$$\text{where } c_{ij} = \left\| \mathbf{x}_i^t - \mathbf{y}_j^t \right\|_2^2.$$

8: $\quad$ Obtain analysis probability distribution $p(\mathbf{x}_a) = \sum_{i=1}^M \sum_{j=1}^N u_{ij}^a \delta_{\mathbf{z}_{ij}}$ where $\mathbf{z}_{ij} = \eta \mathbf{x}_i^t + (1-\eta) \mathbf{y}_j^t$.

9: $\quad$ Obtain $M$ analysis ensemble members $\mathbf{x}_{ai} \in \mathbb{R}^m$ by multinomial sampling from $p(\mathbf{x}_a)$.

10: $\quad$ Set $\mathbf{x}_i^t := \mathbf{x}_{ai}$.

11: **end for**

---

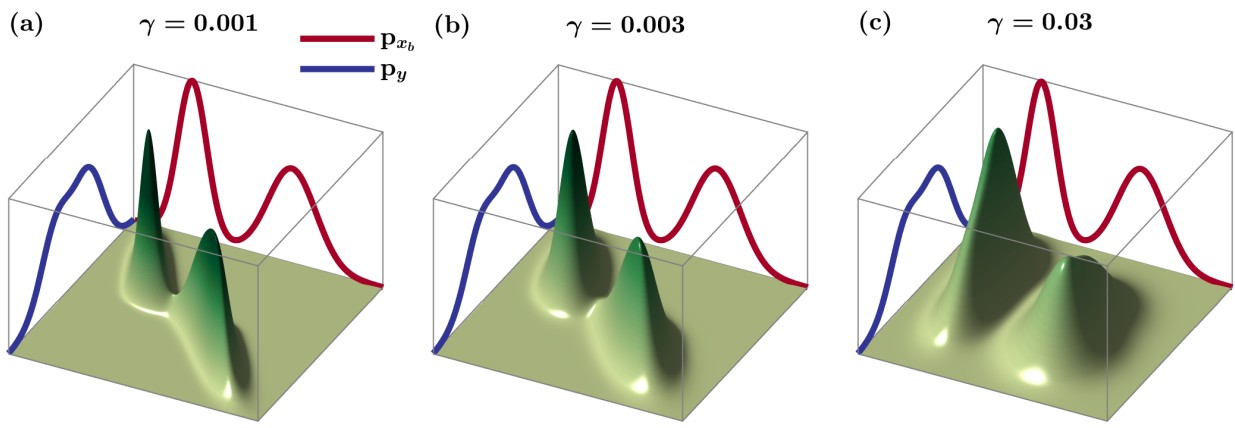

**Figure 3.** The effect of the entropic regularization parameter $\gamma$ on the optimal joint histogram coupling two Gaussian mixture models $\mathbf{p}_{x_b} : 0.5\mathcal{N}(-12, 0.4) + 0.5\mathcal{N}(-8, 0.8)$ and $\mathbf{p}_y : 0.55\mathcal{N}(5, 4) + 0.45\mathcal{N}(9.5, 4)$.

algorithm, assuring sufficient fidelity to the optimal transportation of probability masses according to Eq. (8).





## 4 Numerical Experiments and Results

In order to demonstrate the performance of the EnRDA and quantify its effectiveness, we focus on the
linear advection-diffusion equation and the chaotic Lorenz-63 model (Lorenz, 1963). The advection-diffusion model explains a wide range of heat, mass, and momentum transport across the land, vegetation, and atmospheric continuum, and has been utilized to evaluate the performance of DA methodologies (Zhang et al., 1997; Hurkmans et al., 2006; Ning et al., 2014; Ebtehaj et al., 2014; Berardi et al., 2016). Similarly, the Lorenz-63 model, as a chaotic model of atmospheric convection, has been widely used in testing the performance of DA methodologies (Miller et al., 1994; Nakano et al., 2007; Van Leeuwen, 2010; Goodliff et al., 2015; Tandeo et al., 2015; Tamang et al., 2020). Throughout, under controlled experimental settings with foreknown model and observation errors, we run the forward models under systematic errors and compare the results of the EnRDA with 3D-Var for advection-diffusion dynamics and with the particle filter and EnKF for the Lorenz-63 system.

### 4.1 Advection-Diffusion Equation

#### 4.1.1 State-space Characterization

The advection-diffusion is a special case of the Navier-Stokes partial differential equation. In its linear form, with constant diffusivity in an incompressible fluid flow, it is expressed for a mass conserved physical quantity $\mathbf{x}(\mathbf{s},t)$ as follows:

$$\frac{\partial \mathbf{x}(\mathbf{s},t)}{\partial t} + \mathbf{a} \odot \nabla \mathbf{x}(\mathbf{s},t) = \mathbf{D} \nabla^2 \mathbf{x}(\mathbf{s},t), \tag{14}$$

where $\mathbf{s} \in \mathbb{R}^n$ represents a $n-$dimensional spatial domain at time $t$. In the above expression, $\mathbf{a} = (a_1, \ldots, a_n)^T$ $\in \mathbb{R}^n$ is the advection velocity vector and $\mathbf{D} = \text{diag}(D_1, \ldots, D_n) \in \mathbb{R}^{n \times n}$ represents the diffusivity matrix. Given initial condition $\mathbf{x}(\mathbf{s}, t=0)$, owing to its linearity, the solution at time $t$ can be obtained by convolving the initial condition with a Kronecker delta function $\delta(\mathbf{s} - \mathbf{a}\,t)$ followed by a convolution with the fundamental Gaussian kernel $\mathcal{G}(\mathbf{s},t) = \dfrac{1}{(2\pi)^{n/2} |\mathbf{\Sigma}|^{1/2}} \exp\left(-\dfrac{1}{2} \mathbf{s}^{\mathrm{T}} \mathbf{\Sigma}^{-1} \mathbf{s}\right)$, where $\mathbf{\Sigma} = 2\mathbf{D}\,t$.



### 4.1.2 Experimental Setup and Results

In this subsection, we present the results of DA experiments on 1-D and 2-D advection-diffusion equations. For the 1-D case, the state-space is characterized over a spatial domain $s \in (0, 60]$ with a discretization of $\Delta s = 0.1$. The model parameters are chosen to be $a = 0.8$ [L/T] and $D = 0.25$ [L$^2$/T]. The initial state resembles a bimodal mixture of Gaussian distributions obtained by superposition of two Kronecker delta functions $x(s, t = 0) = 300\,\delta(s)$ – evolved for time 15 and 25 [t], respectively. The ground truth of the trajectory is then obtained by evolving the initial state at a time step of $\Delta t = 0.5$ over a time period of $T = 0$–30 [t] in the absence of any model error.

The observations are obtained at assimilation intervals $10\Delta t$, assuming an identity observation operator, through corrupting the ground truth by a heteroscedastic Gaussian noise with a variance $\epsilon_y = 5\%$ of the squared values of the ground truth state. We introduce both systematic and random errors in model simulations. For the systematic error, model velocity and diffusivity coefficient are set to $a' = 0.12$ [L/T] and $D' = 0.4$ [L$^2$/T] respectively. To impose the random error, a heteroscedastic Gaussian noise with variance $\epsilon_b = 2\%$ is added at every $\Delta t$ to model simulations. One hundred ensembles are used in EnRDA and the regularization and displacement parameters are set to $\gamma = 3$ and $\eta = 0.2$ by trial and error. To obtain a robust conclusion about the comparison of the proposed EnRDA methodology with 3D-Var, experiments are repeated for 50 independent simulation scenarios.

The evolution of the initial state over a time period $T = 0 - 30$ [t] and the results comparing the EnRDA with 3D-Var at 5, 15, and 25 [t] are shown in Fig. 4. As demonstrated, during all time steps, EnRDA reduces the analysis uncertainty, in terms of both bias and unbiased root mean squared error (ubrmse). The shape of the entire state-space is properly preserved and remains closer to the ground truth. As shown, in the 3D-Var, although the analysis state follows the true state reasonably well for initial time steps, as the system propagates over time under systematic errors, the analysis state deviates further away from the ground truth. It is important to note that the displacement parameter in EnRDA is largely determined by the bias while the relative weights in 3D-Var are solely based on the background and observation errors. In particular, under the 3D-Var experiment, the average value of the relative weight we assign to the background state $\alpha = \text{tr}(\mathbf{R})/\text{tr}(\mathbf{R} + \mathbf{B})$ is around 0.4 while in EnRDA this weight is $\eta = 0.2$. Thus in EnRDA, we favored the unbiased observations more and thus the observed improvements in comparison with the 3D-Var might not be completely fair. Because the displacement parameter $\eta$ can be tuned for



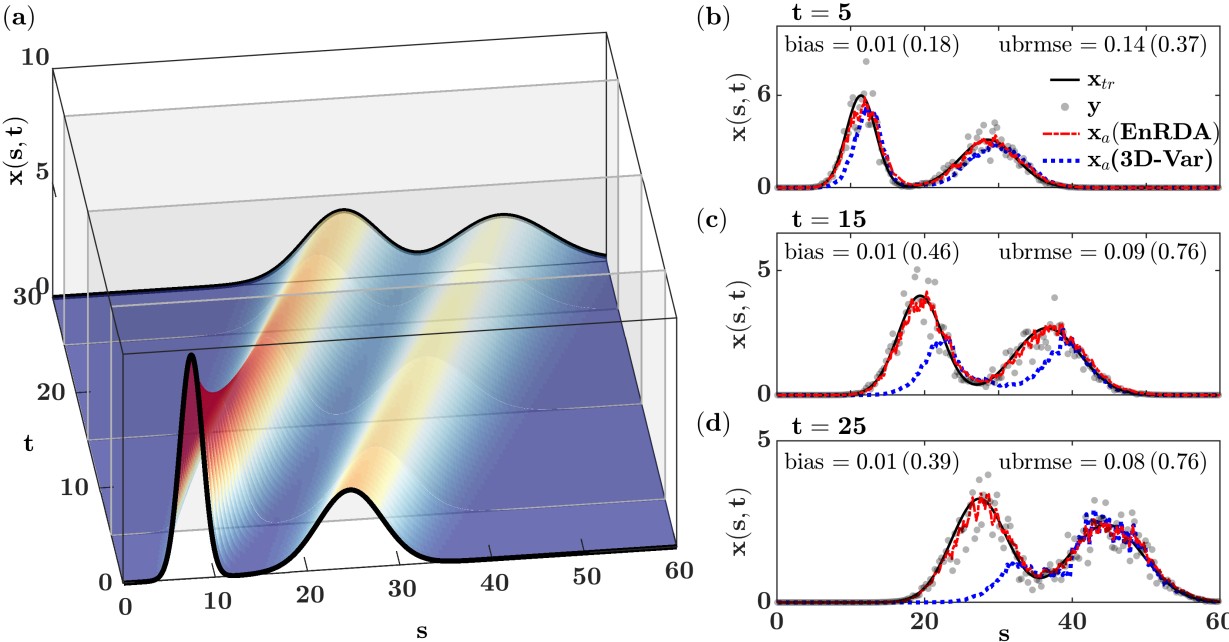

**Figure 4.** (a) Temporal evolution of a bimodal initial state under a linear advection-diffusion equation and (b–d) the true state $\mathbf{x}_{tr}$, observations $\mathbf{y}$ and analysis states $\mathbf{x}_a$ by 3D-Var and EnRDA for three time snapshots at 5, 15 and 25 [t]. The bias and ubrmse of the analysis state by EnRDA (3D-Var) are reported in the legends.

example based on the mean squared error that encompasses the effect of bias but there is no such a mechanism available in 3D-Var. Can EnRDA improve the analysis uncertainty even when $\alpha$ and $\eta$ are comparable?

Fig. 5 shows the results of a 2-D assimilation into the advection-diffusion equation where the underlying state is bimodal. In this example problem, the state-space is characterized over a spatial domain $s_1 = (0, 10]$ and $s_2 = (0, 10]$ with a discretization of $\Delta s_1 = \Delta s_2 = 0.1$. The advection-diffusion is considered to be an isotropic process with the true model parameter values set as $a_1 = a_2 = 0.08$ [L/T], and $D_1 = D_2 = 0.02$ [L²/T]. The shown state variable is obtained after evolving two Kronecker delta functions $\mathbf{x}(\mathbf{s}, t) = 1000\,\delta(s_1, s_2)$ and $\mathbf{x}(\mathbf{s}, t) = 4000\,\delta(s_1, s_2)$ for 25 and 35 [t], respectively.

To resemble a model with systematic errors, background state is obtained by increasing the advective velocity to 0.12 [L/T] while diffusivity is reduced to 0.01 [L²/T] (Fig. 5b). Observations are not considered to have position biases; however, a systematic representative error is imposed assuming that the sensing system has a lower resolution than the model. To that end, we evolve two Kronecker delta functions,




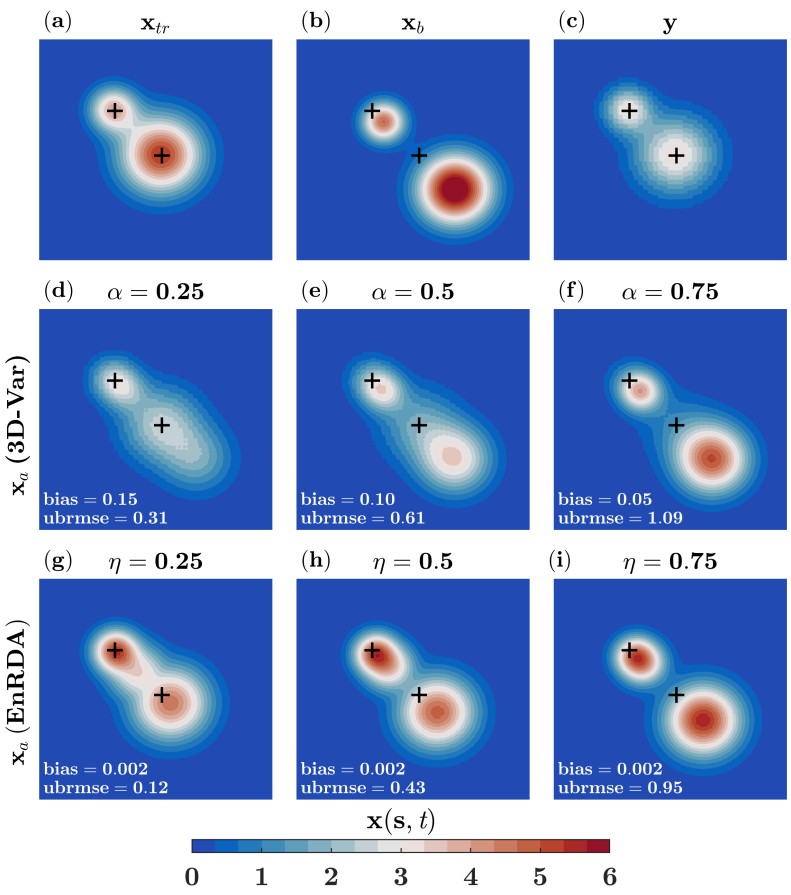

**Figure 5.** The true state $\mathbf{x}_{tr}$ versus the background states $\mathbf{x}_b$ and observations $\mathbf{y}$ (a–c) with systematic errors under a 2-D advection-diffusion dynamics as well as the analysis state $\mathbf{x}_a$ by 3D-Var (d–f) and EnRDA (g–i) for different displacement parameters in the Euclidean $\alpha$ and Riemannian space $\eta$, where the entropic regularization parameter is set to $\gamma = 0.003$. The black plus signs show the location of the modes for the true state.

$\mathbf{x}(\mathbf{s}, t) = 800\, \delta(s_1, s_2)$ and $\mathbf{x}(\mathbf{s}, t) = 2400\, \delta(s_1, s_2)$, with less mass than the true state for same time period of 25 and 35 [t] and then up-scaled the field by a factor of two through box averaging.

As shown in Fig. 5, the EnRDA preserves the shape of the state variable well and gradually moves the mass towards the background state as the value of $\eta$ increases, while the bias remains almost constant and the ubrmse increases from 0.12 to 0.95. The error quality metrics are constantly below the 3D-Var counterpart. The shape of the analysis state for small values of $\alpha$ is not well recovered in 3D-Var due to the position bias. As $\alpha$ increases from 0.25 to 0.75, 3D-Var nudges the analysis state towards the

background state and begins to recover the shape. The bias is reduced by more than 30%, from 0.15 to 0.05; however, this occurs at the expense of almost three folds increase in ubrmse, from 0.3 to 1.1. The




reason for reduction of the bias is that the positive differences between the analysis state and true state are compensated by their negative differences. However, ubrmse is quadratic and thus measures the average magnitude of the error irrespective of its signs. We should emphasize that the presented results do not

imply that EnRDA is always superior to 3D-Var. Indeed, 3D-Var is a minimum mean squared estimator and cannot be outperformed by EnRDA in the absence of bias in a state space with Gaussian distribution.

### 4.2 Lorenz-63

#### 4.2.1 State-space Characterization

The Lorenz system (Lorenz-63, Lorenz, 1963) is derived through truncation of the Fourier series of the

Rayleigh-Bénard convection model. This model can be interpreted as a simplistic local weather system only involving the effect of local shear stress and buoyancy forces. The system is expressed using coupled ordinary differential equations that describe the temporal evolution of three coordinates $x$, $y$, and $z$ representing the rate of convective overturn, horizontal, and vertical temperature variations as:

$$
\begin{aligned}
\frac{dx}{dt} &= -\sigma(x - y) \\
\frac{dy}{dt} &= \rho x - y - xz \\
\frac{dz}{dt} &= xy - \beta z \,,
\end{aligned}
\tag{15}
$$

where $\sigma$ represents the Prandtl number, $\rho$ is a normalized Rayleigh number proportional to the difference in temperature gradient through the depth of the fluid and $\beta$ denotes a horizontal wave number of the convective motion. It is well established that for parameter values of $\sigma = 10$, $\rho = 28$ and $\beta = 8/3$, the system exhibits chaotic behavior with the phase space revolving around two unstable stationary points located at $(\sqrt{\beta(\rho-1)}, \sqrt{\beta(\rho-1)}, \rho-1)$ and $(-\sqrt{\beta(\rho-1)}, -\sqrt{\beta(\rho-1)}, \rho-1)$.

#### 4.2.2 Experimental Setup and Results

In this subsection, we demonstrate the results of DA in the Lorenz system under systematic error using EnRDA, particle filter and EnKF. Throughout, we use the classic multinomial resampling for implementation of EnRDA and particle filter. Apart from the systematic error component, we utilize the standard experimental setting used in numerous DA studies (Miller et al., 1994; Furtado et al., 2008; Van Leeuwen, 2010;





Amezcua et al., 2014). In order to obtain the ground truth of the model trajectory, the system is initialized

at $\mathbf{x}_0 = (1.508870, -1.531271, 25.46091)$ and integrated with a time step of $\Delta t = 0.01$ over a time period

of $T = 0$–$20$ [t] using the fourth-order Runge-Kutta approximation (Runge, 1895; Kutta, 1901). The ob-

servations are obtained at every assimilation interval $40\Delta t$ by assuming identity observation operator and

perturbing the ground truth with Gaussian noise $\boldsymbol{v}_t \sim \mathcal{N}(0, \sigma_{obs}^2 \boldsymbol{\Sigma}_\rho)$, where $\sigma_{obs}^2 = 2$ and the correlation

matrix $\boldsymbol{\Sigma}_\rho \in \mathbb{R}^{3 \times 3}$ is populated with 1 on the diagonal entries, 0.5 on the first sub and super diagonals,

and 0.25 on the second sub and super diagonals.

In order to characterize the distribution of the background state, 100 particles (ensemble members) of

particle filter, EnKF, and EnRDA are generated by corrupting the ground truth at the initial time with a

zero-mean Gaussian noise $\boldsymbol{\omega}_0 \sim \mathcal{N}(0, \sigma_0^2 \mathbf{I}_3)$, where $\sigma_0^2 = 2$. For introducing systematic errors, the model

parameters are set to $\sigma' = 10.5$, $\rho' = 27$, and $\beta' = 10/3$. The random errors are also introduced as the sys-

tem evolves in time by adding a Gaussian noise $\boldsymbol{\omega}_t \sim \mathcal{N}(0, \sigma_b^2 \mathbf{I}_3)$ at every $\Delta t$, with $\sigma_b^2 = 0.02$. Throughout,

to draw a robust statistical conclusion about the error statistics, the DA experiments are repeated for 50

independent simulations. As described previously, to properly account for the effects of both bias and

ubrmse, the optimal value of the displacement parameter $\eta$ in EnRDA can be selected based on an offline

analysis of the minimum mean squared analysis or forecast error. However, to provide a fair comparison

between the EnRDA and other filtering methods, at each assimilation cycle, we set $\eta = \text{tr}(\mathbf{R})/\text{tr}(\mathbf{R} + \mathbf{B})$

assuming that the observation operator is an identity matrix. Note that while the observation error covari-

ance remains constant in time, the background error covariance is obtained from simulated ensembles by

EnRDA and changes in time dynamically. This selection assures that the relative weights assigned to the

background state and observations remain at the same order of magnitude among different methods.

Fig. 6 shows the temporal evolution of the ground truth and the analysis state by the particle filter

(first column), EnKF (second column), and EnRDA (third column) over a time period of 0 to 15 [t] for

one simulation. As is evident, the particle filter is well capable of capturing the ground truth when the

observations lie within the particle spread. However, when the observations lie far apart from the support

set of particles (Fig. 6, dashed box) and the pdfs of the background state and observations become disjoint,

the filter becomes degenerate and the analysis state (particle mean) deviates away from the ground truth. It

is to note that due to the systematic error, the particles in the $z$-coordinate lie away from the observations

and the trajectory fluctuates around the mean of the ground truth (Fig. 6 g, dashed box). As a result, the

bias of the particle filter along the $z$-dimension is markedly lower than that of the EnKF and the EnRDA





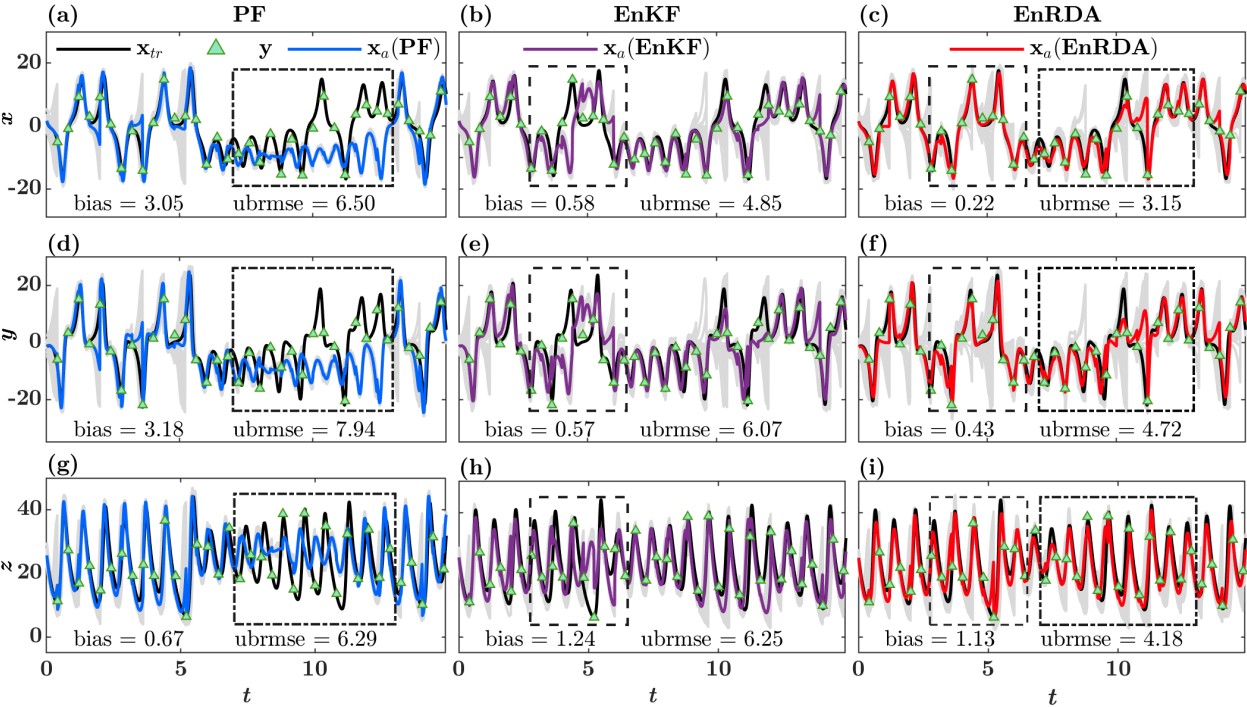

**Figure 6.** Temporal evolution of the true state $\mathbf{x}_{tr}$ of the Lorenz-63, observations $\mathbf{y}$ as well as the analysis state $\mathbf{x}_a$ for the particle filter (PF) (first column), EnKF (EnKF) (second column) and EnRDA (EnRDA) (third column) with 100 particles (ensemble members) respectively. The temporal evolution of the particles and ensemble members are shown with solid gray lines. Also shown within dashed rectangles are the windows of time over which support sets of observations and particle spread are disjoint in particle filter as well as EnKF and EnRDA deviate from the ground truth.

while ubrmse is significantly higher. Whereas, both EnKF and EnRDA are capable of capturing the true state well even when ensemble spread and observations are far apart from each other. Although EnKF does not suffer from the same problem of filter degeneracy as the particle filter, in earlier time steps from 2.5 to 7.5 [t], it struggles to adequately nudge the analysis state towards the ground truth when ensemble members are far from the observations due to the imposed systematic bias. EnRDA seems to be robust to

the propagation of systematic biases in this region and follows the true trajectory well.

The time evolution of the mean values of the bias and ubrmse for 50 independent simulations, with the same error structure, is color coded over the phase space in Fig. 7. As is evident, these forecast quality metrics are relatively lower for the EnRDA than the EnKF and particle filter throughout the simulation period. Nevertheless, we can see that the improvement compared to the EnKF is modest. In particular,

across all dimensions of the problem, the mean bias and ubrmse are decreased in EnRDA by 68 (13)% and 53 (27)% compared to the particle filter (EnKF). More detailed information about the expected values of





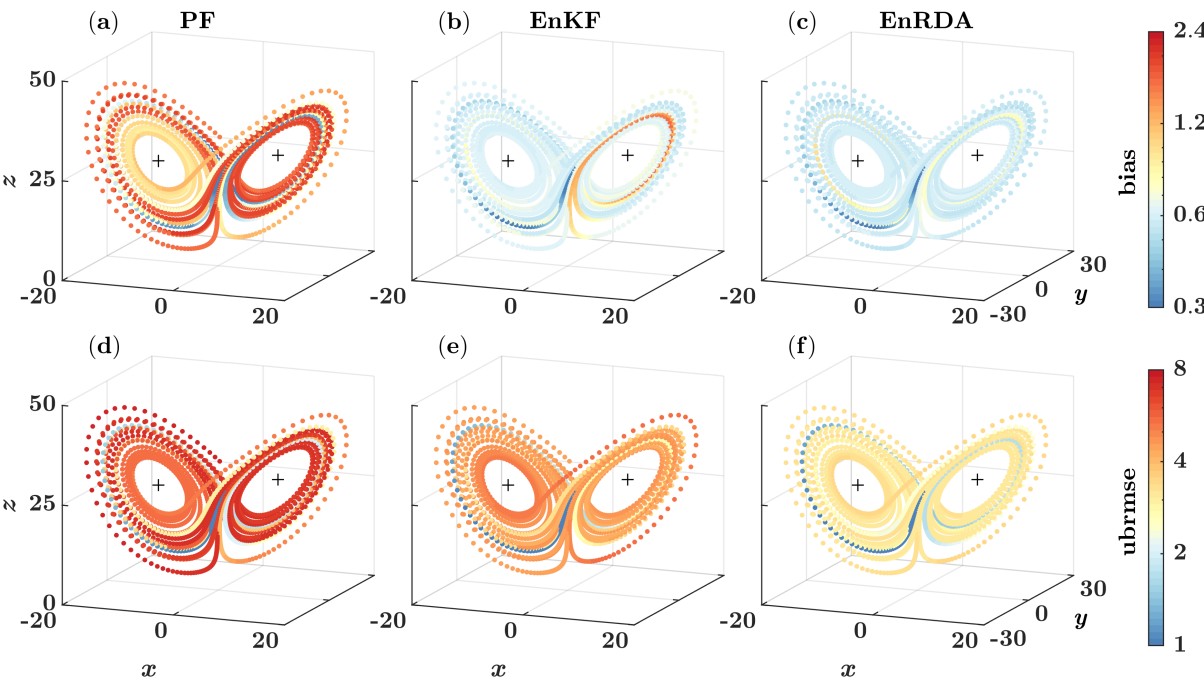

**Figure 7.** Temporal evolution of bias and ubrmse along three dimensions of the Lorenz-63 for (a, d) particle filter, (b, e) EnKF, and (c, f) EnRDA, each with 100 particles (ensemble members). The mean values are computed over 50 independent simulations.

**Table 1.** Expected values of the bias and ubrmse for the particle filter, EnKF and EnRDA from 50 independent simulations of Lorenz-63 across all problem dimensions.

| Methods | bias | | | | ubrmse | | | |
|---|---|---|---|---|---|---|---|---|
| | $x$ | $y$ | $z$ | $x-z$ | $x$ | $y$ | $z$ | $x-z$ |
| Particle Filter | 2.24 | 2.41 | 0.59 | 1.75 | 6.25 | 7.95 | 7.88 | 7.36 |
| EnKF | 0.33 | 0.35 | 1.23 | 0.64 | 3.80 | 5.41 | 5.02 | 4.74 |
| EnRDA | 0.17 | 0.24 | 1.25 | 0.56 | 2.63 | 4.0 | 3.78 | 3.47 |

the bias and ubrmse are reported in Table 1. We emphasize that the presented results shall be interpreted in light of the presence of systematic biases. In fact, EnRDA cannot reduce the analysis error variance beyond a minimum mean squared estimator such as EnKF in the absence of biases.

## 5 Discussion and Concluding Remarks


In this study, we introduced an ensemble data assimilation (DA) methodology over a Riemannian manifold, namely Ensemble Riemannian DA (EnRDA), and illustrated its performance in comparison with a





few Euclidean DA techniques for dissipative and chaotic dynamics.. We demonstrated that the presented

methodology is capable of assimilating information in probability domain – characterized by the families

of distributions with finite second-order moments. The key message is that when the probability distri-

bution of the forecast and observations exhibit non-Gaussian structure and their support sets are disjoint,

due to the presence of systematic errors; the Wasserstein metric can be leveraged to potentially extend

geophysical forecast skills. Even though, future research for a comprehensive comparison with existing

filtering and bias correction methodologies is needed to completely characterize relative pros and cons

of the proposed approach – especially when it comes to the ensemble size and optimal selection of the

displacement parameter $\eta$.

We explained the role of regularization and displacement parameter in EnRDA and empirically exam-

ined their effects on the optimal joint histogram, coupling the background state and observations, and

consequently on the analysis state. Nevertheless, future studies are required to characterize closed-form

or heuristic expressions to expand our understating of their impacts on the forecast uncertainty. As was

explained earlier, unlike the Euclidean DA methodologies that assimilate available information using

different relative weights across multiple dimensions through the error covariance matrices; a scalar dis-

placement parameter is utilized in the the EnRDA that interpolates uniformly between all dimensions of

the problem. Future research can be devoted to developing a framework that utilizes a vector represen-

tation of the displacement parameters to effectively tackle possible heterogeneity of uncertainty across

multiple dimensions.

In it's current form, the EnRDA requires the observation operator to be smooth and bijectve. This is a

limitation when observations of all problem dimensions are not available and propagation of observations

to non-observed dimensions is desired. Extending the EnRDA methodology to include partially observed

systems seems to be an important future research area. This could include performing a rough inversion

for unobserved components of the system offline or extending the methodology in the direction of particle

flows (Van Leeuwen et al., 2019).

Lastly, we should mention that the EnRDA is computationally expensive as it involves estimation of the

coupling through the Wasserstein distance. On a desktop machine with a 3.4 GHz CPU clock rate, it took

around $1600\,\mathrm{s}$ to complete 50 independent simulations on Lorenz-63 for the EnRDA compared to 651

(590) s for the particle filter (EnKF) with 100 particles (ensemble members). Since the computational cost

is nonlinearly related to the problem dimension, it is expected that it grows significantly for large-scale



geophysical DA and becomes a limiting factor. Although the entropic regularization works well for the presented low dimensional problems, future research is needed to test its efficiency in high-dimensional problems. Constraining the solution of the coupling on a submanifold of probability distributions with a Gaussian mixture structure (Chen et al., 2019b) can be a future research direction for lowering the computational cost.

*Code availability.* A demo code for EnRDA in the MATLAB programming language can be downloaded at https://github.com/tamangsk/EnRDA

*Author contributions.* S.K.T. and A.E. designed the study. S.K.T. implemented the formulation and analyzed the results. P.J.L. and G.L. provided conceptual advice and all authors contributed to the writing.

*Competing interests.* The authors declare no competing interests.

*Acknowledgements.* The first and second author acknowledge the grant from the National Aeronautics and Space Administration (NASA) Terrestrial Hydrology Program (THP, 80NSSC18K1528) and the New (Early Career) Investigator Program (NIP, 80NSSC18K0742). The third author acknowledges support from the European Research Council for funding via the Horizon2020 CUNDA project under number 694509. The fifth author also acknowledges support from National Science Foundation (NSF, DMS1830418).





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
