# Peer review of "Ensemble Riemannian Data Assimilation over the Wasserstein Space"

_Nonlinear Processes in Geophysics, 2021_

## Author Comment (AC1)

**Ensemble Riemannian Data Assimilation over the Wasserstein Space**

By: Sagar K. Tamang, Ardeshir Ebtehaj, Peter J. van Leeuwen, Dongmian Zou, and Gilad Lerman

**Responses to review comments (Reviewer 1)**

Comment: The authors present and apply a new methodology for data assimilation that includes non-Euclidean metric and has a potential for addressing systematic and random errors. Results from applications to linear advection-diffusion equation and to Lorenz-63 model show that the new system is performing better than several referent DA systems. A limited discussion of computational cost and applicability of the new method is also included. The methodology and mathematical formalism are sound, but the choice of experiments and presentation can be improved to highlight the pros and cons of the new method. I recommend the manuscript to be accepted subject to major changes. My comments are listed below.

Reply: We very much appreciate the reviewer for providing us with constructive feedback. Your feedback helped us to improve the manuscript. In the revised manuscript with the track-change on, the blue colored text is the updated text in response to the reviewer's comments. Some of the changed text in the manuscript are copied in the replies and shown with red color for convenience. Please also find item-by-item replies to the comments as follows.

General Comments:

1. The inclusion of 3D-Var in the suite of experiments is not clearly justified. 3D-Var is a deterministic/variational method, while the new method is ensemble-based. It is already understood how ensemble and variational methods work and differ. In addition 3D-Var is not very much competitive with Ensemble Kalman Filter (EnKF) or Particle Filter (PF) methods anyway, so showing that the new method outperforms 3D-Var is not saying much. If there is a desire to compare the new method with deterministic methods, then using 4D-Var instead of 3D-Var would be more advisable. Please justify using 3D-Var or exclude it from the experiments.

   Reply: We completely agree with the reviewer's comment that the ensemble-based methodologies are known to perform better than 3D-Var. We admit that the text was confusing. When we structure the manuscript, we first define 3D-Var data assimilation and interpret it as an Euclidean barycenter problem. Then, we evolve the concept of barycenter to the Wasserstein space to define the presented methodology. Therefore, we felt that the initial comparisons between the Euclidean and Wasserstein barycenter makes sense and highlight the differences between data assimilation in the two spaces – particularly in the presence of bias.

   Indeed, Figure 5 (old revision) is just a comparison between the Euclidean and Wasserstein barycenter problems. We considered that 3D-Var and the Euclidean barycenter problem are equivalent, which is correct, but confusing in the way it was presented in Figures 4 and 5. To improve the readership of the manuscript, we have now swapped

the position of Figures 4 and 5 and replaced 3D-Var implementation in Figure 4 (old version) with EnKF and particle filter. Therefore, in this new revision, Figure 4 is just a comparison between the Euclidean and Wasserstein barycenter problem under systematic error to inform the reader about their differences. Figure 5 then shows the comparison between the presented methodology and well-known particle filter and ensemble Kalman filter. The 3D-Var implementation is completely removed from the paper and Figure 5 (revised) is updated accordingly. Please see lines 288–290, 292–293, and 330–339, where we responded to this feedback in the text.

2. The linear advection-diffusion and Lorenz-63 applications include different sets of data assimilation (DA) systems: 3D-Var is used in advection-diffusion and EnKF and PF are used in Lorenz-63 model. This does not allow clear understanding how the new system performs relative to other DA algorithm and what is its true capability in these different applications (e.g., linear/diffusion and nonlinear/chaotic). Please include EnKF and PF in linear advection-diffusion application, and 3D-Var in Lorenz-63 application (conditional on the comment-1 above).

   Reply: Thank you for this comment. As mentioned in the reply to previous comment, in response we decided to completely remove 3D-Var from our analyses and comparisons. In the revision, the results of DA experiments on 1-D advection-diffusion only compares the presented methodology with EnKF and PF. Please see the revised text in lines 330–339 and updated Figure 5.

3. There is no clear understanding if the new method is better suited for handling systematic or random errors. Please add the standalone experiments: (i) random error only and (ii) systematic error only, using the same setup as in the random + systematic error experiment. This will help understand better the true potential of the new system.

   Reply: We appreciate the feedback. Data assimilation over the Wasserstein space is suited for handling systematic errors and might not outperform, in terms of root mean squared error (rmse), the minimum variance unbiased estimators such as EnKF when we only deal with zero mean random errors in linear problems. Please see the text in lines 340–345 (also copied below for convenience), where we have highlighted this issues.

   We should emphasize that the presented results do not imply that EnRDA always performs better than PF and EnKF. The EnKF at the limiting case $M \to \infty$, in the absence of bias, is a minimum mean squared error estimator and attains the lowest possible posterior variance for linear systems, also referred to as the Cramer-Rao lower bound (Cramér, 1999; Rao et al., 1973). Thus, when the errors are drawn from zero-mean Gaussian distributions with a linear observation operator, EnKF can outperform EnRDA in terms of the mean squared error.

Specific comments:

1. One should be aware that systematic errors can be addressed within variational and ensemble methods. Your comparison includes other DA methodologies without that capability. Could you please briefly comment on that in the manuscript?

Reply: Thank you for this comment. Please see the text in lines 413–416 and 434–438 (also copied below for convenience), where we have updated the text in response to this comment.

"Even though, future research for a comprehensive comparison with existing filtering and bias correction methodologies is needed to completely characterize relative pros and cons of the proposed approach – especially when it comes to the ensemble size and optimal selection of the displacement parameter $\eta$."

"Furthermore, it is important to note that several bias correction methodologies are available that explicity add a bias term to the control vector in variational and filtering DA techniques (Dee, 2003; Reichle and Koster, 2004; De Lannoy et al., 2007). Future research is required to compare the performance of EnRDA with other bias correction methodologies to fully characterize its relative advantages and disadvantages."

2. Abstract includes "Eulerian penalization of error in Euclidean space". However this is not a common terminology which requires some explanation (as done in main text). Please adjust Abstract to avoid this terminology until explained.

Reply: Thank you. Please see the updated abstract in lines 2–4. That narrative is removed with simpler explanations.

3. p.1-3: In general, the manuscript includes new terminology that I would not consider common. I would prefer to see some basic explanation of uncommon terminology (such as Wasserstein metric/space, Riemannian manifold, entropic regularization, Sinkhorn algorithm), to allow the reader to go through the manuscript without the need to immediately check the references.

Reply: Thank you. In this round of revision, we have tried our best to avoid the use of uncommon terminologies prior to their explanation. We also modified the sentences in lines 48–49 and 53–56, according to the suggestion as below. Please see text where we explained the Wasserstein distance (line 54–56), entropic regularization (lines 223–228) and Sinkhorn's algorithm (lines 243–247).

"Non-Gaussian statistical models often form geometrical manifolds, a topological space that is locally Euclidean."

"To answer this question, inspired by the theories of optimal mass transport (Villani, 2003), this paper presents the Ensemble Riemannian Data Assimilation (EnRDA) framework using the Wasserstein metric, which is a distance function defined between probability distributions, as explained in more detail in Section 2.3"

4. p.5, L.111-115: I am not sure that 3D-Var the authors refer to is the same as the commonly used 3D-Var. Aside from using a pre-assigned static background error covariance matrix, standard 3D-Var is not including the full matrix inversion that allows direct solution of the linear problem (e.g., in Eq.(26) of Lorenc (1986) the matrix $\mathbf{B}\mathbf{H}^{\mathrm{T}}\mathbf{R}^{-1}\mathbf{H} = 0$). Please refrain from using 3D-Var to describe the system you use, or include some clarification.

Reply: We are referring to the most basic formulation of 3D-Var, based on explanation in the book titled "Atmospheric Modeling, Data Assimilation and Predictability" (Page

168) by Eugenia Kalnay. In response to the earlier comment, we have now removed 3D-Var from our analyses and comparisons with EnRDA.

5. p.6, L.146: Is the cost with subscript "q" and superscript "q" a typo? Can they differ?

Reply: They are the same. The definition of the $\ell_q$-norm is presented in the notation section (line 86). The norm here is just the $\ell_q$-norm to the power $q$. In practice, Euclidean distance is the most common choice in which $q = 2$ and defines the 2-Wasserstein distance.

6. p.8, L178-180: Can this be explained in somewhat more detail? Otherwise, not clear why Cholesky decomposition is relevant here.

Thank you for this feedback! We agree that the text was not complete. We have revised the text in lines 179–185 (also copied below for your convenience).

"The 3D-Var problem in Eq. (2) reduces to the above barycenter problem when the model and observation error covariances are diagonal with uniform error variances across multiple dimensions of the state-space. For non-diagonal error covariances, it can be shown that the weight of the background and observation are $(\mathbf{B}^{-1} + \mathbf{H}^{\mathrm{T}}\mathbf{R}^{-1}\mathbf{H})^{-1}\mathbf{B}^{-1}$ and $(\mathbf{B}^{-1} + \mathbf{H}^{\mathrm{T}}\mathbf{R}^{-1}\mathbf{H})^{-1}\mathbf{H}^{\mathrm{T}}\mathbf{R}^{-1}$ respectively, where $\mathbf{H}$ is the linear approximation of the observation operator. Therefore, the 3D-Var DA can be interpreted as a "barycenter problem" in the Euclidean space, where the analysis state is the weighted mean of the background state and observation."

7. p.11, Eq.(13): How is assured that Kw, Kv are not zero, which is required for the element-wise ratio?

Reply: Thank you. To initialize Sinkhorn's algorithm, we set $\mathbf{w}^0 = \mathbb{1}_N$. The multiplication of the Gibbs Kernel with $\mathbf{v}$ and $\mathbf{w}$ will be zero if and only if the marginal densities in Eq. (13) are zero vectors, which is not the case in this study. Please see lines 246–247, where we clarified this issue.

8. p.13, Eq.(14): A vector dot-product is typically used for advection term. Is this a typo or you are using an element-wise product? Please explain.

Reply: We are using the notation $\odot$ to represent the Hadamard element-wise product between equal length vectors. This notation is described in Section 2.1. Please see lines 87–88 in the revised manuscript.

9. p.13, L. 290: How difficult would be to obtain the optimal parameters eta in realistic high-dimensional situation?

Reply: Thank you for this comment. Similar to the low-dimensional problems, the optimal interpolation parameter $\eta$ can be obtained using the minimum mean-squared error criterion in high-dimensional settings. However, given the high dimensionality of the problem, the computational cost of cross validation can be cumbersome. This issue is addressed in lines 444–445.

We would like to take this opportunity and once gain thank you for taking the time and providing us a thorough review of the manuscript. We did our best to incorporate your

comments into the new revision. We hope that the replies and changes we made in the manuscript meet your expectation.

**References**

Cramér, H.: Mathematical methods of statistics, vol. 9, Princeton university press, 1999.

De Lannoy, G. J., Reichle, R. H., Houser, P. R., Pauwels, V., and Verhoest, N. E.: Correcting for forecast bias in soil moisture assimilation with the ensemble Kalman filter, Water Resources Research, 43, 2007.

Dee, D. P.: Detection and correction of model bias during data assimilation, Meteorological Training Course Lecture Series (ECMWF), 2003.

Rao, C. R., Rao, C. R., Statistiker, M., Rao, C. R., and Rao, C. R.: Linear statistical inference and its applications, vol. 2, Wiley New York, 1973.

Reichle, R. H. and Koster, R. D.: Bias reduction in short records of satellite soil moisture, Geophysical Research Letters, 31, 2004.

Villani, C.: Topics in optimal transportation, 58, American Mathematical Soc., 2003.

---

## Author Comment (AC2)

**Ensemble Riemannian Data Assimilation over the Wasserstein Space**

By: Sagar K. Tamang, Ardeshir Ebtehaj, Peter J. van Leeuwen, Dongmian Zou, and Gilad Lerman

**Responses to review comments (Reviewer 2)**

Comment: This paper introduces an optimal transport framework for updating discrete representations of posterior probability density functions during ensemble data assimilation. This work further provides proof-of-concept assimilation experiments comparing the algorithm the authors introduce (and call EnRDA) to standard DA methods (3D-var, particle filters, ensemble Kalman filters). The authors mention two serious issues that will need to be overcome for EnRDA to become a viable strategy for users in the DA community: 1) the high computational expense (that scales super-cubicly with the ensemble size) associated with computing optimal transport maps and 2) the need for the observation operator to be bijective, e.g. all state dimensions must be observable.

Reply: We very much appreciate the thoughtful review and comments on the manuscript. We have included a track-change color coded version in which red colored text is the updated text in response to the reviewer's feedback. Some of the changed text in the manuscript are also copied in the replies and shown with red color for convenience. Please also find item-by-item replies to your comments as follows.

Some minor comments:

1. The reader would benefit from more discussion of the Sinkhorn algorithm and how/why gamma is chosen as well as how eta is chosen in practice. On line 290, you mention that these parameters are set by "trial and error". Can you offer the reader more guidance on how to make these choices in practice, even if ad-hoc? Do these or should these parameters vary at different assimilation time steps? On line 290 you have gamma=3 and in the caption of Figure 5 you have gamma=0.003. Are both correct? Are these for the same DA experiment at different times? Or different experiments? This should be clarified in the text. In Figure 3 you demonstrate that one order of magnitude change in gamma results in quite different joint distributions with the same prior and observation pdfs. What does a 3 order of magnitude change in gamma do?

   Reply: Thank you for this comment. The displacement parameter $\eta$ can be tuned offline through cross-validation by minimizing the mean squared error or any other error metric of interest. The error shall be defined with respect to a reference point such as ground-based observations. Please see the updated text in lines 196–199, where we have addressed this comment. However, the value of the regularization parameter is highly dependent on the transportation cost matrix. In practice, one can begin with $\gamma$ set as the largest element of the transportation cost matrix and gradually reduce it to find the minimum value of $\gamma$ that provides a stable solution of Sinkhorn's algorithm. Please see the updated text in lines 260–264. Both the displacement and regularization parameters are static in our implementation, however, future research may come up

with new ideas for dynamic updating. Please see line 417–426, where we addressed this issue.

Thanks for your attention to the details. We double checked and the reported values of $\gamma$ are correct. The choice of $\gamma$ is different for different experiments and dependent on the transportation cost matrix. The value of $\gamma = 0.003$ is set for two arbitrary Gaussian mixture models defined in Figure 3 whereas $\gamma = 3$ is set for our experimental setting in the 1-D advection-diffusion model. Please see the updated text in lines 260–264, where we clarified that the value of the regularization parameter is different for different experiments.

2. The barriers to wide usage of this approach are quite high, yet if overcome the EnRDA could prove a very powerful DA method. As such I believe these barriers and the research advances needed to overcome them warrant a longer discussion than you offer in section 5.

Reply: Thank you for your perspective. We agree and did our best to be upfront about the barriers of the presented approach. Please see the expanded discussion about the limitation in Section 5, 432–434, and 449–452 (also copied below for convenience).

"Another promising area is to utilize EnRDA only over the observed dimensions of the state-space and similar to the EnKF, use the ensemble covariance to update the unobserved part of state-space through a hybrid approach."

"Furthermore, recent advances in approximation of the Wasserstein distance using a combination of 1-D Radon projections and dimensionality reduction (Meng et al., 2019), can significantly reduce the computational cost to make EnRDA a viable methodology for tackling high-dimensional geophysical DA problems."

3. Typographical mistake — you have two periods ending a sentence on line 398.

Reply: Thank you, fixed.

Once again we would like to take the opportunity and thank you for the insights and feedback that helped us to improve the manuscript. We hope that the replies and changes we made in the manuscript meet your expectation.

**References**

Meng, C., Ke, Y., Zhang, J., Zhang, M., Zhong, W., and Ma, P.: Large-scale optimal transport map estimation using projection pursuit, in: Advances in Neural Information Processing Systems, edited by Wallach, H., Larochelle, H., Beygelzimer, A., d'Alché-Buc, F., Fox, E., and Garnett, R., vol. 32, Curran Associates, Inc., 2019.